# Low Body Mass Index as a Predictive Factor for Postoperative Infectious Complications after Ureterorenoscopic Lithotripsy

**DOI:** 10.3390/medicina57101100

**Published:** 2021-10-13

**Authors:** Kensaku Seike, Takashi Ishida, Tomoki Taniguchi, Shota Fujimoto, Daiki Kato, Manabu Takai, Koji Iinuma, Keita Nakane, Hiromi Uno, Masayoshi Tamaki, Hisao Komeda, Takuya Koie

**Affiliations:** 1Department of Urology, Chuno Kosei Hospital, Seki 5013802, Japan; k-seike@chuno.gfkosei.or.jp (K.S.); t-tani@chuno.gfkosei.or.jp (T.T.); h-uno@chuno.gfkosei.or.jp (H.U.); 2Department of Urology, Gifu Municipal Hospital, Gifu 5008513, Japan; justaskaxis@gmail.com (T.I.); mtamaki@gmhosp.gifu.gifu.jp (M.T.); hkome@gmhosp.gifu.gifu.jp (H.K.); 3Department of Urology, Ogaki Municipal Hospital, Ogaki 5038502, Japan; uro2@omh.ogaki.gifu.jp; 4Department of Urology, Gifu University Graduate School of Medicine, Gifu 5011194, Japan; andreas7@gifu-u.ac.jp (D.K.); takai_mb@gifu-u.ac.jp (M.T.); kiinuma@gifu-u.ac.jp (K.I.); keitaco@gifu-u.ac.jp (K.N.)

**Keywords:** stone disease, ureteroscopy, ureterorenoscopic lithotripsy, infectious complication, body mass index

## Abstract

*Background and Objectives*: In this study, we aimed to evaluate predictive factors of postoperative fever (POF) after ureterorenoscopic lithotripsy (URSL). *Materials and Methods:* A total of 594 consecutive patients who underwent URSL for urinary stone disease at Gifu Municipal Hospital and Chuno Kosei Hospital between April 2016 and January 2021 were enrolled in this study. In all patients, antibiotics were routinely administered intraoperatively and the next day after surgery. We used rigid and/or flexible ureterorenoscopes depending on the stone location for URSL. Stones were fragmented using a holmium: YAG laser. The fragments of the stone were manually removed as much as possible using a stone basket catheter. A ureteral stent was placed at the end of the surgery in all cases. *Results:* The median age and body mass index (BMI) in all patients were 62 years and 23.8 kg/m^2^, respectively. The median operation duration was 52 min. The most common URSL-related complication was POF in 28 (4.7%) patients. In these patients, the rates of antibiotic administration and ureteral stent insertion before surgery were significantly higher than in those without POF. In multivariate analysis, BMI was associated with POF after URSL. There were no significant differences in predicting POF after surgery in patients who had bacteriuria or received antibiotics before surgery. *Conclusions:* A low BMI was significantly associated with POF after URS or URSL.

## 1. Introduction

Urinary stones are one of the most common benign urologic diseases, with a lifetime incidence of approximately 10% [1], and a common cause of morbidity and deterioration of quality of life worldwide [2]. In Japan, the incidence of lower urinary tract stones has gradually increased from 4.7/100,000 in 1965 to 9.1/100,000 in 2005 [3]. Likewise, the estimated age-standardized annual incidence of upper urinary tract stones was 54.2/100,000 in 1965 and 114.3/100,000 in 2005 [4].

The management of urolithiasis has dramatically changed over the last three decades [5]. Minimally invasive techniques such as ureterorenoscopic lithotripsy (URSL), shockwave lithotripsy (SWL), and percutaneous nephrolithotomy (PNL) are the standard treatment modalities based on patient preference, symptoms, stone location, and stone size [6]. Recently, URSL has been accepted as the first-line treatment choice for ureter and renal stones, with better stone-free rates than SWL and lower complication rates than PNL [7]. However, postoperative complications occur in 2.5–6.7% patients after ureterorenoscopy (URS) [8,9].

Postoperative fever (POF) is the most common complication of ureteroscopic holmium laser lithotripsy, with an incidence rate of 2–28% [8,9,10]. In addition, POF is potentially serious because it may progress to urosepsis, leading to death [11]. Although predictive factors that may be associated with infectious complications in PNL and SWL have been evaluated, only a limited number of studies have identified predictive factors for POF after URSL [12,13]. Furthermore, there is no consensus on the predictive factors that may lead to infectious complications after URS. Therefore, this study aimed to evaluate the predictive factors of POF after URSL.

## 2. Materials and Methods

### 2.1. Patients

The study protocol was approved by the Institutional Review Board of Chuno Kosei Hospital (number: R3-1) and Gifu Municipal Hospital (number: 707). We retrospectively reviewed the data of 630 consecutive patients who underwent URSL for urinary stone disease at Gifu Municipal Hospital and Chuno Kosei Hospital between April 2016 and January 2021. Preoperative data included age, gender, body mass index (BMI), preoperative bacteriuria, history of diabetes mellitus or hypertension, preoperative antibiotic use, and laboratory parameters related to systemic inflammation. The location (ureter or kidney), size, multiplicity of the stones, preoperative ureteral stent insertion, and operation duration were also evaluated. The comorbidities of the enrolled patients using Charlson comorbidity index or the American Society of Anesthesiologists physical status were not evaluated in this retrospective study. Patients with fever (>38 °C) persisting for 48 h after URSL were considered to have POF.

### 2.2. URSL Technique

The procedure was performed under general or spinal anesthesia with the patient in the lithotomy position. All URSLs were performed by three expert surgeons (K.S., T.T., and S.F.). In all patients, antibiotics (usually cefazolin) were routinely administered intraoperatively and the next day after surgery. We used a 6.5/7.5-Fr rigid URS (r-URS) (Richard Wolf Medical Instruments Cooperation, Knittlingen, Germany) or flexible URSs (f-URS), including 7.5-Fr Flex-X2™ (Karl Storz, Tuttlingen, Germany), 9.9-Fr Cobra-M™ (Richard Wolf Medical Instruments Corporation, Knittlingen, Germany), 8.5/9.9-Fr URF-type V (Olympus, Tokyo, Japan), or 4.9/7.95-Fr URF-P7 (Olympus, Tokyo, Japan). 

The first step was to observe the ureter using a r-URS after the insertion of a safety guidewire. URSL was performed using the r-URS for direct access to the target stone. If the target stone could not be observed using the r-URS, an 11/13-12/14-Fr Navigator™ (Boston Scientific, Boston, MA, USA) or 11/13-12/14-Fr Uropass^®^ (Olympus, Tokyo, Japan) was inserted into the ureter for ureteral access. Then, a f-URS was inserted into the ureter and stones were fragmented using holmium:YAG-Laser (Lumenis, Versa Pulse Select 80 W, Yokneam, Israel) and a 200-μm laser fiber with an energy of 0.5–1.0 J and a rate of 5–10 Hz. The SAPS™ CF irrigation system (Boston Scientific, Marlborough, MA, USA) was used to avoid prolonged high-pressure irrigation. The fragments of the stone were picked out as much as possible using a 2.2-Fr NCircle^®^ (COOK, Bloomington, IL, USA), 1.9-Fr. ZeroTip™ (Boston Scientific, Marlborough, MA, USA), or 1.9-Fr Flex Catch (Olympus, Tokyo, Japan). 

A 4.8- or 6-Fr ureteral stent was placed at the end of the surgery in all cases. The ureteral stent was removed 1–2 weeks after URSL.

### 2.3. Statistical Analysis

The primary endpoint was to determine the predictive factors for POF after URS. Data were analyzed using SPSS software (version 24.0; IBM Corp., Armonk, NY, USA). Continuous and categorical variables were compared using the Kruskal–Wallis test. Multivariate logistic regression analysis was performed to evaluate the predictors of POF after URSL. Two-sided *p*-values were calculated, and the significance level was set at *p* < 0.05.

## 3. Results

A total of 594 patients were enrolled in the study. We excluded 36 patients, including 20 patients with bilateral stones who underwent bilateral URSL with one-step surgery and 16 patients with missing data. The median age and BMI in all patients were 62 years (interquartile range [IQR], 50.3–71 years) and 23.8 kg/m^2^ (IQR, 21.3–26.6 kg/m^2^), respectively. The median operation duration was 52 min (IQR, 35–77 min). The most common URSL-related complication was POF in 28 (4.7%) patients. Of these patients, two (0.3%) patients developed sepsis. Subsequently, the surgery-related complications were ureteral injury in eight (1.3%) patients, post-dural puncture headache in six (1.0%) patients, and hypotension in one (0.2%) patient. The commonly reported complications not related to URSL were sinus tachycardia in one (0.2%) patient, arteriosclerosis obliterans in one (0.2%) patient, and transient ischemic attack in one (0.2%) patient.

The demographic data of patients who were classified into two groups according to POF after URSL are listed in Table 1. In patients with POF after URSL, the rates of antibiotic administration and ureteral stent insertion before surgery were significantly higher than in those without POF.

In multivariate analysis, BMI was associated with POF after URSL (Table 2). The incidence of POF after surgery was not significantly different between patients with bacteriuria and those who received antibiotics before surgery.

## 4. Discussion

POF is the most common complication after endoscopic procedures of the urinary tract [14]. To date, many studies have reported the risk factors of POF after URS or URSL [15,16,17,18]. In multivariate analyses, POF after URS or URLS has been shown to be associated with longer operative duration [5,15,16], female patients [15,17,18], preoperative bacteriuria or pyelonephritis [16,17,18], and infectious stones [16,18]. However, several studies have not been able to identify predictive factors for infectious complications after URS or URSL [11,13]. Chugh et al. reported that antibiotic prophylaxis was practiced in most of the included studies, which reduced the risk of infection [19]. Infectious complications vary and include fever, urinary tract infection (UTI), pyelonephritis, systemic inflammatory response syndrome, and urosepsis [19]. Therefore, antibiotics should be tailored to local resistance profiles, which tend to reduce the rates of infection and urosepsis [20]. In addition, several factors, such as preoperative UTI, higher Charlson comorbidity index, elderly or female patients, indwelling ureter stent, operation duration, and patients with high BMI, may be associated with POF [19]. In addition, URSL has many potential causes, including ureteral obstruction by stone fragments, UTI, intraoperative backflow and extravasation of the urine due to prolonged high-pressure irrigation, intraoperative and postoperative bleeding, and postoperative backflow of the urine in the bladder and ureter due to poor drainage via the catheter [21]. In a study by Sugihara et al., severe adverse events were associated with longer operative times and lower hospital volumes [15]. In our study, preoperative bacteriuria, administration of antibiotics before surgery, and C-reactive protein (CRP) levels were not associated with POF in multivariate analysis. Further, the operation duration in this study was similar to that reported in previous studies [5,13,17,18,19]. Therefore, intraoperative backflow or high-pressure irrigation may be a potential risk factor for POF after URSL.

In our study, low BMI was a predictive factor for POF after URSL. To our knowledge, the association between POF after URSL and BMI is unclear. Obesity is a risk factor for the development of kidney disease [22]. In a previous study, the urinary excretion of risk factors for stone formation and inhibitory substances was significantly higher in patients with overweight and obesity than in patients with normal weight and underweight [22]. Vale et al. reported the most (60.6%) patients who were overweight or obese and that an increase in BMI was associated with higher urinary calcium excretion [23]. Moreover, a recent systematic review and meta-analysis has demonstrated a significant association between urolithiasis and metabolic syndrome [24]. However, patients with overweight or obesity did not demonstrate an increased risk according to mortality from sepsis in another study [25]. In a retrospective study using the CERNER™ HealthFacts electronic health record database, Pepper et al. identified 55,038 patients with sepsis between 2009 and 2015 [26]. The BMI groups were underweight (BMI: <18.5 kg/m^2^), normal weight (BMI: 18.5–24.9 kg/m^2^), overweight (BMI: 25.0–29.9 kg/m^2^), and obese (BMI: >30 kg/m^2^) [26]. Using multivariate analysis, the adjusted odds ratio of short-term mortality (death or hospice) was 1.62 for an underweight BMI, 0.73 for an overweight BMI, and 0.61 for an obese BMI [26]. Similarly, the 28-day mortality risk was 1.8-fold higher in the underweight group than in the normal weight group in the overall cohort and 2.9-fold higher in the sepsis sub-cohort [26]. In addition, patients who were underweight had a longer intensive care unit length of stay, increased need for mechanical ventilation support, and a higher frequency of fluid overload [25]. Morokuma et al. have suggested that low BMI may be a risk factor of POF after URSL, despite the small number of patients enrolled in this study [27]. Therefore, patients with a low BMI may develop POF easily after URS or URSL.

Our study has several limitations. First, this was a retrospective study that was conducted using multicenter data. Therefore, this study has an inherent potential for bias, with therapeutic variations among these institutions. Second, a relatively small number of patients were enrolled, and patients with bilateral stones were excluded from this study. Finally, not all patients were examined for urine culture and neutrophil and lymphocyte counts before surgery and analysis of stone components after surgery.

## 5. Conclusions

POF is the most common complication after URS or URSL. In this study, it was not necessary to administer antibiotic prophylaxis, even if the patients had bacteriuria before surgery. In addition, a low BMI was significantly associated with POF after URS or URSL. Therefore, patients with a low BMI need to be more careful with POF such as urosepsis during URS or URSL.

## Figures and Tables

**Table 1 medicina-57-01100-t001:** Comparison of perioperative covariates and outcomes in patients who had postoperative fever after ureteroscopic lithotripsy or not.

Covariates	Patients without POF (*n* = 559)	Patients with POF (*n* = 35)	*p* Value
Age (years, median, IQR)	61 (50–70)	71 (58–81)	0.003
Gender (number, %)			0.001
Male	370 (66.2)	13 (37.1)	
Female	189 (33.8)	22 (62.9)	
Body mass Index (kg/m^2^, median, IQR)	23.9 (21.5–26.8)	21.2 (19.0–24.3)	0.002
Bacteriuria (number, %)			0.473
Negative	131 (23.4)	4 (20.0)	
Positive	145 (26.0)	21 (51.4)	
Unknown	283 (50.6)	10 (28.6)	
Antibiotics use before surgery (number, %)	33 (5.9)	7 (20)	<0.001
Diabetes Mellitus (number, %)	79 (14.1)	8 (22.9)	0.157
Hypertension (number, %)	224 (40.1)	10 (28.6)	0.177
Stone location (number, %)			0.023
Kidney	58 (10.4)	8 (22.9)	
Ureter	501 (89.6)	27 (72.1)	
Stone size (mm, median, IQR)	9.7 (7.0–12.0)	10.0 (7.3–14.9)	0.229
Number of stones			0.345
1	443 (75.7)	24 (68.6)	
≥2	136 (24.3)	11 (31.4)	
Indwelling of ureter stent before surgery (number, %)	154 (27.5)	20 (57.1)	<0.001
Initial laboratory data			
Leucocyte count (/μL, median, IQR)	6290 (5105–7630)	6400 (5050–7425)	0.760
Platelet count (×10^2^/μL, median, IQR)	23.5 (19.8–27.8)	22.9 (16.7–28.0)	0.276
CRP (mg/dL, median, IQR)	0.18 (0.05–0.67)	0.57 (0.01–1.90)	0.017
Operation duration (minutes, median, IQR)	52 (35.3–75.8)	66 (31.5–101.5)	0.140
Residual stone (number, %)	58 (10.4)	3 (8.6)	0.733

POF, postoperative fever; IQR, interquartile range; CRP, C-reactive protein.

**Table 2 medicina-57-01100-t002:** Multivariate logistic regression test according to postoperative fever after ureteroscopic lithotripsy.

Risk Factors	*p* Value	Odds Ratio	95% CI
Body mass index	0.042	0.903	0.819–0.996
Stone location	0.053	2.412	0.990–5.873
Gender	0.068	0.495	0.232–1.054
CRP	0.089	1.094	0.986–1.214
Indwelling ureter stent before surgery	0.115	0.530	0.240–1.167
Age	0.200	1.018	0.991–1.045
Antibiotics use before surgery	0.317	0.601	0.222–1.628

CI, confidence interval; CRP, C-reactive protein.

## Data Availability

Data and materials are provided in this paper.

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
