# Peer review of "Low Body Mass Index as a Predictive Factor for Postoperative Infectious Complications after Ureterorenoscopic Lithotripsy"

_medicina, 2021, doi:10.3390/medicina57101100_

Round 1

Reviewer 1 Report

This paper relates a retrospective study of patients undergoing ureteroscopic removal of stones, with details on postoperative complications. I have some questions and suggestions for the authors:

  1. The number of patients in the study and reasons for exclusion need some amplification, I think. The number for analysis was 594, but 630 cases were reviewed. Why were patients with ‘bilateral stones’ excluded? Did this include persons who had ureteroscopy on only one side but also had a stone on the other? Or was this an exclusion of patients who were treated on both sides in the same operation?
  2. Abstract, lines 19-20: This is an awkward sentence. As I understand the methods, you used a rigid scope to remove some ureteral stones, but a flexible scope was used for renal access. This sentence in the Abstract suggests that you sometimes used a rigid scope to treat stones in the kidney.
  3. Abstract, line 21: I think just ‘using a basket.’ A ‘catheter’ generally implies a tubular object with a lumen.
  4. You might consider adding the word ‘low’ to the title.

Author Response

11, Oct, 2021

Dr. Miljan Petrovic

Assigned Editor

Medicina

Dear Editor:

Thank you very much for the review of our manuscript titled “Low body mass index as a predictive factor for postoperative infectious complications after ureterorenoscopic lithotripsy”

We sincerely appreciate all valuable comments and suggestions, which helped us to improve the quality of our manuscript. Our responses to the Reviewers’ comments are described below in a point-to-point manner. Appropriate changes, suggested by the Reviewers, have been introduced to the manuscript (track-changes mode in the red color font). Let me emphasize our full readiness to make any further improvements to the manuscript.

We hope that our manuscript will be acceptable for publication in the Medicina.

We look forward to hearing from you.

Yours sincerely,

Takuya Koie

Department of Urology

Gifu University Graduate School of Medicine

1-1 Yanagido, Gifu, Gifu 501-1194, Japan

TEL.: +81-582-30-6338

FAX: +81-582-30-6341

Responses to the reviewer's comments

We would like to thank the Reviewers for taking the time and effort necessary to review the manuscript. We sincerely appreciate all the valuable comments and suggestions, which helped us to improve the quality of the manuscript.

Response to Reviewer 1

The authors appreciate the reviewer’s comments. The authors’ point-by-point responses to the comments are given below.

  1. The number of patients in the study and reasons for exclusion need some amplification, I think. The number for analysis was 594, but 630 cases were reviewed. Why were patients with ‘bilateral stones’ excluded? Did this include persons who had ureteroscopy on only one side but also had a stone on the other? Or was this an exclusion of patients who were treated on both sides in the same operation?

Response:

We have added the following sentence, on line 99:

We excluded 36 patients, including 20 patients with bilateral stones who underwent bilateral URSL on one-step surgery and 16 patients with missing data.

  1. Abstract, lines 19-20: This is an awkward sentence. As I understand the methods, you used a rigid scope to remove some ureteral stones, but a flexible scope was used for renal access. This sentence in the Abstract suggests that you sometimes used a rigid scope to treat stones in the kidney.

Response:

We have added the following sentence, on line 19:

We used rigid and/or flexible ureterorenoscopes depending on the stone location for URSL.

  1. Abstract, line 21: I think just ‘using a basket.’ A ‘catheter’ generally implies a tubular object with a lumen.

Response:

We have added the following sentence, on line 21:

The fragments of the stone were manually removed as much as possible using a stone basket catheter.

 You might consider adding the word ‘low’ to the title.

 Response:

The authors have revised the title according to the reviewer’s recommendation as follows.

Low body mass index as a predictive factor for postoperative infectious complications after ureterorenoscopic lithotripsy

Reviewer 2 Report

The aim of the study was to evaluate predictive factors of POF after URSL. The study is interesting and properly structured. There are just a few minimal aspects that Authors need to review in order to improve overall quality of manuscript.  Authors should report comorbidites of patients (Charlson comorbidity index) and ASA score, in order to better compare baseline characteristics.

Why stone location (p=0.023) was not included in the mulitivariate logistic regression?

Conclusion must be modified in order to highlight results obtained from the current study, particularly underlining its impact on clinical practice. 

Author Response

11, Oct, 2021

Dr. Miljan Petrovic

Assigned Editor

Medicina

Dear Editor:

Thank you very much for the review of our manuscript titled “Low body mass index as a predictive factor for postoperative infectious complications after ureterorenoscopic lithotripsy”

 We sincerely appreciate all valuable comments and suggestions, which helped us to improve the quality of our manuscript. Our responses to the Reviewers’ comments are described below in a point-to-point manner. Appropriate changes, suggested by the Reviewers, have been introduced to the manuscript (track-changes mode in the red color font). Let me emphasize our full readiness to make any further improvements to the manuscript.

We hope that our manuscript will be acceptable for publication in the Medicina.

We look forward to hearing from you.

Yours sincerely,

Takuya Koie

Department of Urology

Gifu University Graduate School of Medicine

1-1 Yanagido, Gifu, Gifu 501-1194, Japan

TEL.: +81-582-30-6338

FAX: +81-582-30-6341

Responses to the reviewer's comments

We would like to thank the Reviewers for taking the time and effort necessary to review the manuscript. We sincerely appreciate all the valuable comments and suggestions, which helped us to improve the quality of the manuscript.

Response to Reviewer 2

The authors appreciate the reviewer’s comments. The authors’ point-by-point responses to the comments are given below.

The aim of the study was to evaluate predictive factors of POF after URSL. The study is interesting and properly structured. There are just a few minimal aspects that Authors need to review in order to improve overall quality of manuscript.  Authors should report comorbidites of patients (Charlson comorbidity index) and ASA score, in order to better compare baseline characteristics.

Why stone location (p=0.023) was not included in the mulitivariate logistic regression?

Conclusion must be modified in order to highlight results obtained from the current study, particularly underlining its impact on clinical practice. 

 Response:

Unfortunately, we did not evaluate comorbidity index before URSL. We cannot reinvestigate this factor because this study was a retrospective manner.

Therefore, we have added the following sentence, on line 66:

The comorbidities of the enrolled patients using Charlson comorbidity index or the American Society of Anesthesiologists physical status were not evaluated in this retrospective study.

 We have added the stone location as a risk factor according to POF in the Table 2.

 The authors have revised the following sentences, on line 180:

In addition, a low BMI was significantly associated with POF after URS or URSL. Therefore, URS or URSL for patients with a low BMI need to be more careful with POF such as urosepsis.